# Detecting Environmental Stress In Situ Using Molecular Data: A Case Study with the Filamentous Green Alga *Klebsormidium* and Antarctic Biocrusts

**DOI:** 10.3390/microorganisms13092108

**Published:** 2025-09-09

**Authors:** Deepamalini Palaniappan, Ekaterina Pushkareva, Burkhard Becker

**Affiliations:** 1Department of Plant Science, IZMB Institute, University of Bonn, 53115 Bonn, Germany; malinipalani98@gmail.com; 2Department of Biology, Institute for Plant Sciences, University of Cologne, 50923 Cologne, Germany; ekaterina.pushkareva@uni-koeln.de

**Keywords:** desiccation stress, cold stress, metatranscriptomics, marker genes, gene marker indices

## Abstract

The polar environment is one of the most extreme environments of our world. However, even in the cold deserts of Antarctica, life thrives, often in the form of biocrusts (biological soil crusts)—complex communities consisting of hundreds of organisms. The reaction to abiotic stress in members of these communities is often inferred from laboratory experiments on isolated species and single factors, without taking into consideration any mitigation effects by the communities or complex habitats. In this study, we aimed to infer the stress situation of the filamentous green alga *Klebsormidium* in Antarctic biocrusts in situ using metatranscriptomic data. *Klebsormidium* is ubiquitous in biocrusts and well studied with respect to abiotic factors, allowing the comparison of lab experiments with the in situ situation. In this study, we identified *Klebsormidium flaccidum* to be present in biocrusts from Livingston Island (Antarctica). Metatranscriptomic data for the biocrust were used to investigate the presence of cold and desiccation stress in situ. To this end, we identified consistently expressed and stress-regulated genes in published stress transcriptomes of *Klebsormidium* that could serve as markers for environmental stress levels. These “marker genes” were used to construct marker gene indices to assess stress states in biocrusts by comparing transcript expression ratios under different conditions—a novel framework for the assessment of microbial community responses to environmental stressors. However, many potential marker genes behaved quite differently in the laboratory and in the natural environment. In the end, rather than relying on indices based on individual marker genes, comparing the expression levels of whole stressor-regulated gene sets proved to be a more reliable approach to examining stress in situ. This study highlights the potential of marker genes for broader ecological and environmental monitoring using metatranscriptomic data.

## 1. Introduction

Antarctica represents one of the most extreme environments on Earth, characterized by persistent cold, strong winds, high levels of UV radiation, and seasonal desiccation [1]. Despite these extreme conditions, the continent’s ice-free areas, confined to coastal margins [2], support unique ecosystems that are crucial in understanding biological adaptations to harsh environments [1]. The organisms inhabiting these zones have adapted to extreme conditions, including low temperatures (mean temperatures ranging from −10 °C to −60 °C), high solar radiation, and limited nutrients. The terrestrial microbiota in Antarctica face challenges such as desiccation and osmotic stress, yet they have developed strategies to thrive in these conditions. The availability of liquid water is a pivotal factor in the growth and development of the terrestrial microbiota, but it is only available during the short austral summer [3]. Diverse microbial communities are found in various habitats, including soils, biocrusts, rocks, mats, and sediments. Nevertheless, the composition of these communities varies greatly between different localities, even over short distances [4,5].

Biocrusts are defined as assemblies of organisms that colonize the soil [6,7]. In comparison with underlying soils, biocrusts have a higher abundance and diversity of microorganisms. These microorganisms are crucial for carbon and nutrient cycling in Antarctica, where only two vascular plant species are present [6]. The bacterial communities in Antarctic soils and biocrusts are predominantly composed of Actinobacteria, Acidobacteria, and Bacteroidetes, with many other less abundant phyla [5,8,9,10].

The genus *Klebsormidium,* a member of the Streptophyta, has been found in a variety of habitats worldwide [11] and frequently occurs in polar biocrusts [12,13,14]. *Klebsormidium* is a filamentous alga that produces exopolysaccharides, which contribute to soil stabilization and microbial interactions within the biocrusts [15,16]. Polar strains of *Klebsormidium* have been extensively studied for their stress tolerance mechanisms [14,17,18,19]. As poikilohydric organisms, these algae can survive extreme dehydration, relying on desiccation tolerance mechanisms that allow for rapid recovery upon rehydration [20].

The extreme environment of Antarctica subjects its inhabitants to multiple abiotic stresses, including sub-zero temperatures, desiccation, high salinity, and intense UV radiation. Numerous laboratory experiments have demonstrated that *Klebsormidium* has evolved a range of protective strategies, such as the accumulation of cryoprotectants and the upregulation of antioxidant defense mechanisms [14,15,16,17,19,21,22,23,24,25]. However, the specific stress responses of *Klebsormidium* species within Antarctic biocrusts under natural habitats remain underexplored. Gaining a better understanding of how these microorganisms endure the multifaceted stresses of their environment in situ will provide valuable insights into the resilience, ecological roles, and functional dynamics of Antarctic biocrusts.

The objective of this study was to address the knowledge gap between laboratory experiments and the in situ conditions by evaluating the stress situations of *Klebsormidium* strains in Antarctic biocrusts. To achieve this, stress-related marker genes suitable for assessing the physiological states of *Klebsormidium* strains in their natural environment were identified. The expression levels of these genes in in situ samples were compared with those from laboratory-based stress experiments [19] to determine transcripts that are constantly expressed, upregulated, or downregulated in response to stress. By analyzing the expression patterns for these genes, the proportions of regulated to constantly expressed genes could be calculated, providing insights into the stress status of *Klebsormidium* within its natural habitat.

## 2. Materials and Methods

### 2.1. Site Description and Sampling

The sampling sites were located on the Byers Peninsula in Livingston Island, Antarctica. The peninsula is the largest ice-free area in Maritime Antarctica and features over 60 freshwater lakes and streams. The mean air and soil (at 5 cm depth) temperatures throughout the year are approximately −2.7 °C and −1.3 °C, respectively [26]. Three sites were selected at the Byers Peninsula (Table 1, Figure 1). At each site, five field replicates were collected and stored at +4° C. For metatranscriptomic analysis, additional samples were immediately placed into 2 mL tubes containing LifeGuard^®^ Soil Preservation Solution (QIAGEN, Hilden, Germany) to maintain RNA integrity and kept at −20 °C until RNA extraction. The environmental parameters of the camp at the time of sampling were recorded by the Agencia Estatal de Meteorologia (AEMET) (Madrid, Spain) for the Byers Field station (Table 1).

### 2.2. Isolation and Cultivation of Klebsormidium from Livingston Island

To provide experimental evidence for the presence of *Klebsormidium* strains within the collected biocrusts, microalgae were isolated from enrichment cultures. Single colonies were transferred to Petri dishes with Bold’s basal medium triple-nitrogen concentration (3N-BBM) and added vitamins. The cultures were kept at 20 °C under 30 mol photons m^−2^ s^−1^ (Osram Lumilux Cool White lamps L36W/840, OSRAM GmbH, Munich, Germany) with a light/dark regime of 16/8 h. To identify the species, samples were taken from the cultures and examined using a Keyence Biozero microscope (Frankfurt, Germany) at magnifications ranging from 20× to 60×. Images were taken for documentation purposes. The species were identified using the relevant literature [27].

### 2.3. RNA Isolation and Sequencing

To perform RNA extraction, the frozen cryotubes containing the biocrust samples were gradually thawed and centrifuged at 2500× *g* for 5 min, and the soil preservation solution was removed. Total RNA was then extracted from each sample using the RNA Soil Mini Kit (MACHEREY-NAGEL, Düren, Germany), according to the manufacturer’s instructions. Metatranscriptomic sequencing with poly(A) enrichment (PE150) was conducted at the Cologne Center for Genomics (CCG, Cologne, Germany) using the Illumina MiSeq platform (Illumina, San Diego, CA, USA). Due to the low quality of the RNA in some samples, only three (site 1) and four replicates (site 3), respectively, were sequenced. The resulting sequences were deposited in the Sequence Read Archive (SRA) under the project number PRJNA1263187.

### 2.4. Bioinformatic Analysis

Bioinformatic analysis was performed using the OmicsBox software (v3.0.30) with standard settings, if not indicated otherwise [28]. The quality filtering of the obtained files was performed with Trimmomatic (v0.39) [29], and rRNAs were separated from the dataset using SortMeRNA (v4.3.7) [30]. The remaining reads were assembled de novo using Trinity (v2.15.2) [31] (integrated into OmicsBox), with separate assemblies conducted for each site. Transcript quantification was performed using RSEM (v1.3.3) [32] with the Bowtie2 aligner (v2.5.4) [33]. *Klebsormidium* transcripts within the metatranscriptome assemblies were identified using Local BLASTN (v2.16.0) (e-value 1.0 × 10^−50^) in OmicsBox against a custom database of available *Klebsormidium* sequences (draft genome of *K. nitens* GCA_000708835.1 [34] and the transcriptomes of *K. flaccidum* and *K. dissectum* (BioProject ID PRJNA500592 [19], newly assembled as described below).

### 2.5. Identification of Klebsormidium Species Using Bioinformatic Analysis

To identify the *Klebsormidium* species present in the biocrust samples, 18S rRNA sequences were extracted from the metatranscriptomic dataset, assigned with the Silva database (version 138.2), and assembled de novo using the RNA-Seq De Novo Assembly option in OmicsBox. The assembled rRNAs were then subjected to a BLASTN search against the NCBI database NR2024-7-11, limiting the search to Viridiplantae, to identify the *Klebsormidium* species.

### 2.6. Re-Analysis of RNA-Seq Data from a Stress Experiment Involving K. flaccidum and K. dissectum

The two stress transcriptomes of *K. flaccidum* and *K. dissectum* [19] were downloaded from NCBI BioProject ID PRJNA500592. *K. flaccidum* (strain A1-1a; [35]) was originally isolated from an Antarctic biocrust on Ardley Island, South Shetland Islands, while *K. dissectum* (strain EiE-15a; [12]) was isolated from a biocrust in Svalbard, Norway. De novo assemblies were performed separately for *K. flaccidum* and *K. dissectum* reads, as described above. To assess the completeness of the assembled transcriptomes, a BUSCO analysis, using the Viridiplantae dataset and implemented in Omicsbox, was carried out [36]. Coding regions were predicted using TransDecoder (v5.7.1) [37,38]. Functional annotation of the sequences was conducted using local BLAST searches against the *K. nitens* genome [34]. To further improve the annotation of the transcriptomes, Diamond BLAST using the NCBI non-redundant (NR) database (version 2024-01-09), followed by InterProScan [39] and eggnog mapper [40] in the Omicsbox background, was used.

Transcript quantification was performed by mapping the reads to the assembled transcriptomes using, again, RSEM and Bowtie2 [32,33]. A pairwise differential expression analysis was conducted using edgeR (v 3.20.9) [41].

### 2.7. Gene Marker Indices

To identify transcripts for marker indices, the workflow in Figure 2 was followed. *Klebsormidium* transcripts were identified in metatranscriptomes using local BLASTN searches against the *Klebsormidium* database (see above, Section 2.4). Transcripts with 98% sequence identity or more to *K. flaccidum* or *K. dissectum* transcripts were retained for further analysis. The read abundance for each transcript was recorded, and the *Klebsormidium* origin of the transcript was verified through a BLASTx (v2.16.0) search against the NCBI NR-2024-1-9 database, restricted to Viridiplantae.

The identified *Klebsormidium* transcripts were manually compared with the stress transcriptomes of *K. flaccidum* and *K. dissectum* and grouped according to their regulation in the stress transcriptomes (regulated vs. non-regulated). A marker gene index (GMI) was constructed for each pair of regulated and non-regulated transcripts to quantify the effects of stress on gene expression. The GMI was calculated as a percentage of the maximal effect according to the following equation:GMI_i_ = (X_i_ − X_i_min)/(X_i_max − X_i_min) × 100.(1)

X_i_ is the ratio of the number of reads (fpkm) of a stress-regulated transcript to the number of reads of a non-regulated transcript in the biocrust.

X_i_min is the ratio of the number of reads (fpkm) of stress-regulated transcripts to the number of non-regulated transcripts in the control experiment.

X_i_max is the ratio of the number of reads (fpkm) of a stress-regulated transcript to a non-regulated transcript in the experimental treatment.

Each i represents a specific pair of regulated and non-regulated transcripts.

### 2.8. Statistical Tests of Marker Gene Sets

To compare the expression patterns of marker gene sets across different samples and conditions, the Similarity Profile (SIMPROF) test was performed in R (v4.3.1) using the clustsig (v1.1) and vegan (v2.6-8) packages. Transcript counts were first normalized to the total number of *Klebsomidium* reads in the dataset to calculate relative abundances and a Bray–Curtis dissimilarity matrix was created. Hierarchical clustering was then performed using the Ward.D2 method. Two different datasets were used for this analysis. Dataset 1 included only transcripts used for ratio calculation. These transcripts are regulated by either cold or desiccation stress only. Dataset 2 includes all regulated transcripts identified from BLAST-positive contigs, including those that are regulated by both cold and desiccation stress.

## 3. Results

### 3.1. Identification of Klebsormidium flaccidum in Biocrusts from Livingston Island

Strains morphologically resembling *Klebsormidium* were successfully isolated from biocrusts collected at sites 1 and 3 (Figure 3). An analysis of the rRNA from the metatranscriptomic datasets for sites 1 and 3 confirmed the presence of *Klebsormidium flaccidum* at sites 1 and 3.

### 3.2. Identification of Desiccation- and Cold-Regulated Transcripts in K. dissectum and K. flaccidum

To identify genes that are suitable for use as gene marker indices (GMIs), the stress transcriptomes published by Rippin et al. [19] were re-analyzed. To eliminate discrepancies arising from differing workflows or settings, a complete re-analysis was performed, using a consistent pipeline for both the previously published stress transcriptomes and the newly generated metatranscriptomes. The results of this re-analysis are summarized below.

The de novo-assembled transcriptomes of *K. flaccidum* and *K. dissectum* contained 120,574 and 96,581 transcripts, respectively. Summary data for the assemblies, including BUSCO analysis and open reading frame (ORF) detection, are presented in Appendix A. In brief, both assemblies contained approximately 80% complete ORFs and covered more than 95% of the genome, according to BUSCO analysis. A high duplication rate was observed in both K. *flaccidum* (82%) and *K. dissectum* (88%), which contributed to the high number of transcripts. The assemblies were annotated as described in the Materials and Methods, which yielded 71,658 and 73,763 annotated transcripts for *K. flaccidum* and *K. dissectum*, respectively. Pairwise differential gene expression analysis (fold change > 2, *p* < 0.05, FDR < 0.05) identified a large number of transcripts responsive to both cold and desiccation stress in both *Klebsormidium* species (Table 2). These transcripts represent potential marker genes suitable for the calculation of GMIs.

### 3.3. Metatranscriptomes for In Situ Stress Analysis

The metatranscriptomes were assembled using the same pipeline that had previously been applied to the stress transcriptomes (Table 3). The number of assembled transcripts was higher at site 1 than at site 3. Although a substantial number of transcripts showed similarity to *Klebsormidium* sequences, only a small fraction exhibited sequence identity greater than 98%. A subset of these high-similarity transcripts was selected for further analysis.

The selected transcripts were then categorized as either regulated or constantly expressed genes. Approximately one third of the genes regulated by one stressor were also influenced by the other. For example, at site 1, 22 transcripts were regulated by cold stress and 22 by desiccation stress, and 11 were regulated by both factors. Based on the presence of *K. flaccidum* in the biocrusts, as indicated by the sequence similarity, the differential gene expression catalog for *K. flaccidum* was used for comparison. Transcripts that were differentially expressed in response to cold or desiccation in laboratory experiments were identified at all sampling sites through comparison with *K. flaccidum* stress transcriptomes (Table 4). Both site 1 and site 3 exhibited a similar number of regulated and constantly expressed transcripts in response to these stressors. A complete list of selected transcripts and their functional annotations is presented in Appendix A. Only transcripts specifically regulated by a single stressor in the laboratory experiments were included in the calculation of GMIs.

### 3.4. Evaluation of In Situ Stress State in Biocrusts Using GMI

To evaluate the suitability of GMIs as indicators of abiotic stress, GMIs were calculated for all combinations of regulated and constantly expressed transcripts from sites 1 and 3, as listed in Appendix A. Specifically, 143 combinations were analyzed for cold stress at site 1, 130 for cold stress at site 3, and 96 for desiccation stress at both sites. Prior to GMI calculation, transcript counts were normalized to the total number of reads within each dataset.

An analysis of the obtained GMI values revealed that a substantial proportion extended beyond the expected 0–100% range, indicating that the transcript expression levels may have fallen below the observed laboratory control conditions or exceeded the laboratory stress conditions (Figure 4). The GMI values were highly variable, indicating complex regulatory patterns among transcripts and reflecting diverse stress states. Furthermore, over 50% of the GMIs exhibited either a more pronounced marker gene response or values similar to those observed in laboratory stress experiments. In contrast, nearly 40% of the GMIs were lower than, or comparable to, the laboratory control levels. Excluding GMIs that fell outside the laboratory experiment range did not allow a definitive interpretation. For instance, of the 143 combinations analyzed for cold stress at site 1, only 40 produced GMIs within the expected range, with an average effect size of 53.5%. However, the high standard deviation (36.5%) indicates a wide spectrum of stress intensities, ranging from minimal to severe, comparable to those observed in the cold stress experiment. To assess the reliability of individual transcripts as stress indicators, their frequency of occurrence in GMIs within the expected range (Figure 4) was examined. However, no clear pattern was observed that would support the exclusion of specific transcripts from further analyses, indicating that individual marker genes are not reliable indicators when used in isolation. Consequently, the analysis focused on the overall expression patterns of the *Klebsormidium* transcripts.

### 3.5. Comparison of Overall Gene Expression Profiles to Assess Stress State of Biocrusts

Hierarchical clustering of the gene expression profiles revealed clear separation between cold control and cold stress samples in the laboratory datasets, with minimal variation among replicates (Figure 5). In contrast, field samples from sites 1 and 3 exhibited greater variability. Within the cold stress dataset, field replicates grouped together with the cold stress laboratory samples. Notably, one replicate from both sites exhibited expression patterns that were almost identical to those observed under cold treatment in the dataset containing transcripts regulated exclusively by cold stress. In the broader dataset (see Materials and Methods for the differences between the two datasets), which encompassed all regulated *Klebsormidium* genes, field samples were still grouped with cold-stressed samples, albeit at a greater distance. These results strongly suggest that *Klebsormidium flaccidum* in the biocrusts of Livingston Island experienced cold stress at the time of sampling.

The pattern differed in the case of desiccation stress (Figure 6). In dataset 1, field samples from site 1 clustered together with the hydrated samples from the laboratory experiment. For site 3, half of the field samples grouped together with the hydrated or desiccated samples from the laboratory experiment. In contrast, in the larger dataset 2, which included all regulated transcripts likely derived from *Klebsormidium*, field replicates from both sites formed a separate cluster, distinct from the laboratory experiments. Overall, the analysis indicates greater variability among field replicates with respect to the desiccation response (Figure 6B). Furthermore, the inclusion of genes regulated by both cold and desiccation stress dilutes the desiccation-specific signal, potentially obscuring clear clustering with laboratory treatments.

## 4. Discussion

The presence of *Klebsormidium* in Antarctic biocrusts was confirmed through a combination of methods, including the isolation of algal strains, 18S rRNA analysis, and BLAST-based transcriptome analysis. This comprehensive approach enabled the identification of *K. flaccidum*.

### Identification of Marker Genes

The objective of this study was to identify potential marker genes with stable and regulated expression across all experimental stress conditions in all *Klebsormidium* species commonly used in laboratory stress experiments. To ensure the reliability of candidate transcripts, expression patterns were examined across all experiments, aiming to detect genes with constantly regulated expression. This approach aligns with standard practices in the field of comparative transcriptomics, where analyses initially include multiple species or strains to increase the robustness of data interpretation before narrowing the focus to a specific target organism. This methodology facilitates the identification of conserved and species-specific stress response pathways [42,43]. However, this approach yielded only approximately 20 genes, which were not detected in the metatranscriptomic datasets. Consequently, the analysis was refined to focus on individual stressors and the *K. flaccidum* dataset, with the objective of expanding the number of stress-related transcripts included in the analysis. This targeted approach resulted in the identification of several potential marker genes in the metatranscriptomic dataset.

Gene marker indices (GMIs) represent a novel approach to quantifying environmental stress, providing a framework to assess organismal stress responses and enabling direct comparisons between controlled laboratory conditions and natural environments. The normalization of gene expression to constantly expressed reference genes (commonly referred as housekeeping gene normalization) is a standard practice in transcriptomic studies [44]. This normalization ensures that any observed changes in expression reflect genuine biological responses rather than technical artifacts. Building upon this established methodology, GMIs incorporate normalized expression values into ratio-based metrics, providing an innovative means of evaluating relative stress levels across both experimental and field settings. Despite the absence of directly comparable studies, due to the novelty of the method, related approaches have successfully linked gene expression profiles to environmental stress. For example, stress-responsive microbial genes in biocrusts have demonstrated consistent expression patterns across laboratory and field conditions, highlighting the potential of molecular indicators in ecological monitoring [45].

The tested GMIs exhibited a wide range, indicating no stress to strong stress, suggesting that many of the proteins regulated under laboratory stress conditions might be either subjected to more complex regulation in situ or not be involved in the stress response under natural conditions. This finding might suggest that metatranscriptomic data may not accurately reflect abiotic environmental conditions. However, further analysis of the global transcript expression levels, focusing on transcripts previously identified in stress experiments, revealed a clear correlation with environmental parameters, particularly with cold stress. This correlation is most evident in the context of low temperatures. For example, at the time of sampling, the biocrusts were exposed to soil temperatures below 5 °C. In both datasets and across both investigated sites, the transcript expression patterns closely mirrored the environmental conditions. In the larger dataset, the observed weakening of the clustering might reflect the simultaneous regulation of transcripts by multiple environmental factors, which we observed quite often for cold and desiccation stress. Regulation by multiple stressors could result in the dilution of the signals associated with individual stressors. However, in the context of desiccation stress, a different pattern was observed. In the smaller dataset, only a weak association was recorded with the hydrated control, while, in the larger dataset, comprising transcripts responsive to both desiccation and temperature, the experimental conditions demonstrated greater similarity to each other. Two explanations might be proposed regarding this pattern. Firstly, although the biocrusts appeared dry at the time of sampling, the air humidity was high (Table 1), suggesting that the surface biofilm might have absorbed dew and was, therefore, hydrated. If this was the case, the expression pattern for dataset 1 accurately indicates the hydrated state. The variability among replicates could be attributed to the different proportions of the upper biofilm that were included during RNA extraction. Secondly, previous studies have shown that cold acclimation in *Klebsormidium* can reduce the sensitivity to desiccation [19]. Thus, the expression patterns observed in field samples might be significantly influenced by cold stress, resulting in an increased number of transcripts with different expression patterns compared to the lab desiccation experiment.

Despite the potential of metatranscriptomic data for in situ stress evaluation, their application remains in the early stages and requires further methodological development. One key limitation is the need for comprehensive transcriptomic reference datasets to establish reliable baselines for both control and stress conditions. In addition to temperature and desiccation, other important environmental factors, such as the light intensity (PAR, UV) and nutrient availability, must also be considered in order to fully capture the complexity of natural stress responses. Moreover, the natural variability in environments and the interactions between multiple simultaneous stressors may complicate data interpretation. Future research should thus aim at the automation of the selection of marker genes and the analysis of expression patterns using machine learning algorithms, thus enhancing the reproducibility and scalability.

In conclusion, marker-based metatranscriptome analysis is a powerful tool for the assessment of environmental stress, effectively bridging the gap between controlled laboratory experiments and the complex responses of microbial communities in natural environments. The findings of this study highlight the potential of this method for ecological monitoring, conservation applications, and the broader understanding of microbial adaptation to extreme environments.

## Figures and Tables

**Figure 1 microorganisms-13-02108-f001:**
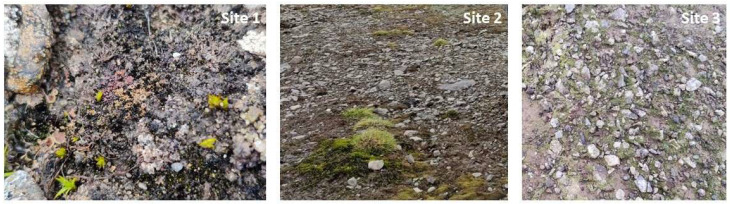
Sampling sites on Livingston Island in Antarctica. The images show the three sampling sites on the Byers Peninsula, where the BSC samples were collected for this study.

**Figure 2 microorganisms-13-02108-f002:**
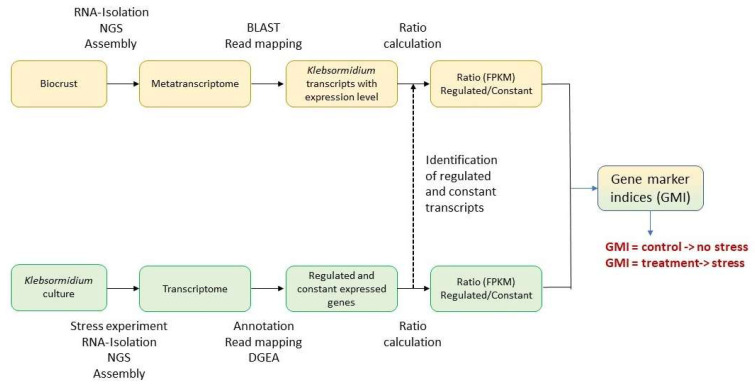
Bioinformatics workflow for detection of in situ stress responses of *Klebsormidium* in biocrust metatranscriptomes.

**Figure 3 microorganisms-13-02108-f003:**
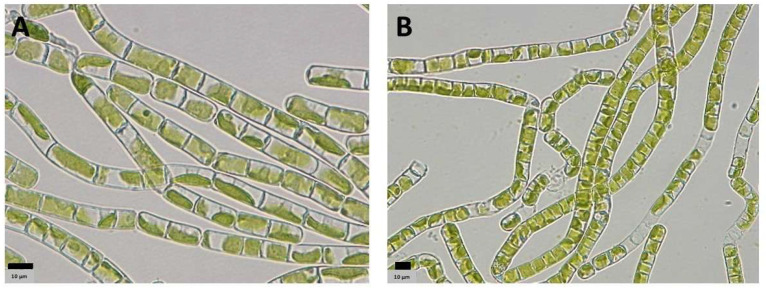
*Klebsormidium flaccidum* strains isolated from sites 1 (**A**) and 3 (**B**). Scale bar = 10 µm.

**Figure 4 microorganisms-13-02108-f004:**
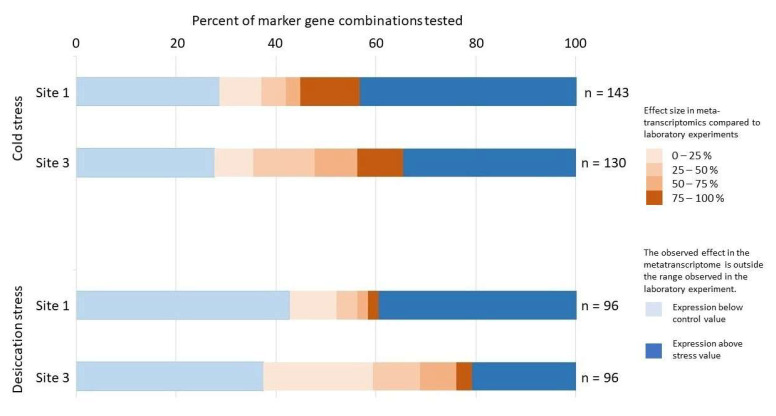
The GMI distribution from the metatranscriptomic field data is compared to that in the laboratory stress and control conditions. The percentage of GMI combinations outside the observed effect range in the laboratory experiment is shown in blue. The percentage of GMI combinations within the effect range of the stress experiments is shown in brown, with different shades depicting different effect strengths. Most GMI combinations are outside the control or treatment range, indicating that they are regulated differently than in laboratory experiments.

**Figure 5 microorganisms-13-02108-f005:**
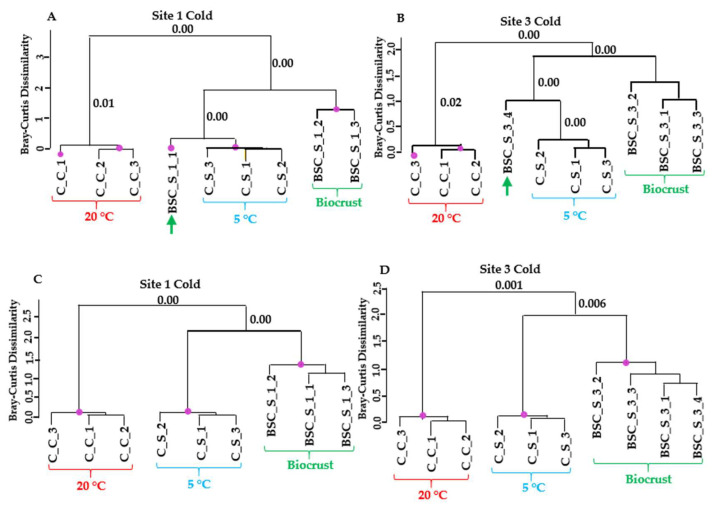
Similarity in transcript expression patterns between laboratory cold treatment of *K. flaccidum* and biocrust samples from sites 1 and 3. (**A**,**B**) Dendrograms based on regulated transcripts used for GMI calculation. (**C**,**D**) Dendrograms based on all regulated transcripts identified as likely derived from *Klebsormidium*. Purple dots indicate statistically significant clusters based on SIMPROF analysis using Bray–Curtis dissimilarity. The *p*-values corresponding to the merging of significant clusters are indicated near the respective branches in the dendrograms.

**Figure 6 microorganisms-13-02108-f006:**
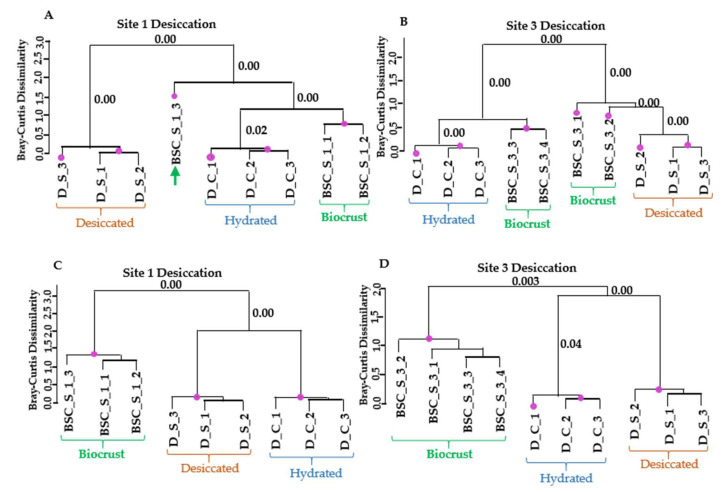
Similarity in transcript expression patterns between laboratory desiccation treatment of *K. flaccidum* and biocrust samples from sites 1 and 3 from Livingston Island. (**A**,**B**) Dendrograms based on regulated transcripts used for GMI calculation. (**C**,**D**) Dendrograms based on all regulated transcripts identified as likely derived from *Klebsormidium*. The purple dots indicate statistically significant clusters based on SIMPROF analysis using Bray–Curtis dissimilarity. The *p*-values corresponding to the merging of significant clusters are indicated near the respective branches in the dendrograms.

**Table 1 microorganisms-13-02108-t001:** Characteristics of the investigated sites. The location, sampling date, and sampling site, as well as the air and soil temperatures, are given. The macroscopic appearance is also indicated.

	Site 1	Site 2	Site 3
Location	62°39′39.4847″S 61°5′46.9031″ W	62°39′55.785″ S 61°6′02.052″ W	62°39′50.5″ S, 61°06′01.6″ W
Sampling date	9 January 2023	9 January 2023	11 January 2023
Sampling time	15:55:00	17:07:00	18:50:00
Characteristics	Biocrust dominated by lichens and moss	Algae-dominated biocrust; 100–150 m away from petrel nests and dead seals	Algae-dominated biocrust; 30 m away from last tent in camp, mild human disturbance
Soil temperature	4.6 °C	3.8 °C	2.8 °C
Air temperature	2.3 °C	2.2 °C	2.0 °C
Relative humidity	99.9%	100%	99.8%

**Table 2 microorganisms-13-02108-t002:** Number of transcripts regulated in laboratory stress experiments in *K. flaccidum* and *K. dissectum.* The total number of upregulated and downregulated genes is given for cold and desiccation treatment.

Stress	No. of Transcripts Regulated in *K. flaccidum*		No. of Transcripts Regulated in *K. dissectum*	
	Upregulated	Downregulated	Upregulated	Downregulated
Cold stress	8576	12,725	8004	7556
Desiccation	4435	5034	1897	3957

**Table 3 microorganisms-13-02108-t003:** Metatranscriptomic analyses were performed on poly(A)-enriched RNA isolated from biocrust at sites 1 and 3. A summary of the key characteristics of the de novo metatranscriptome assemblies for sites 1 and site 3, as well as the number of BLAST hits for *Klebsormidium* and the number of transcripts used in further analyses, is provided.

		Site 1	Site 3
Raw reads	Bases before processing	7,825,615,239	11,906,629,319
De novo metatranscriptome assembly	No. transcripts	679,102	300,904
	No. genes	534,509	227,696
	N50 [bases]	491	736
	Hits with *Klebsormidium* BLAST library	46,341	23,618
BLAST analysis	Similarity > 98%, e-value < 1 × 10^−66^	123	81
Manual selection	Selected transcripts used for GMIs	35	31

**Table 4 microorganisms-13-02108-t004:** *Klebsormidium* transcripts identified in Antarctic metatranscriptomes and their corresponding expression profiles in the stress transcriptomes of *K. flaccidum* in laboratory experiments.

	Expression Profile in *K. flaccidum* Stress Transcriptome	Site 1	Site 3
Effect of cold	Constantly expressed transcripts	13	13
	Regulated transcripts	11	10
Effect of desiccation	Constantly expressed transcripts	12	12
	Regulated transcripts	8	8

## Data Availability

The reads for the metatranscriptomic dataset have been submitted to the SRA archive at the NCBI under PRJNA1263187.

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
