# Peer review of "Detecting Environmental Stress In Situ Using Molecular Data: A Case Study with the Filamentous Green Alga Klebsormidium and Antarctic Biocrusts"

_microorganisms, 2025, doi:10.3390/microorganisms13092108_

Round 1

Reviewer 1 Report

Comments and Suggestions for Authors

I enjoy reviewing the MS about the detection of environmental stress in Klebsormidium using a metatranscriptomic approach. I agree with the authors that Klebsormidium is one of the most widely distributed genera with high tolerance to extreme ecological factors. I am sure that the reviewed article will be of interest to a wide range of scientists. In the study, the authors used proper molecular-genetic and statistical methods. All parts of the MS are clear, and the data are presented logically. It is necessary to note that the paper is well edited. The results are illustrated by clear figures. The conclusions of the paper are based on the results. The supplementary file contains useful information about Klebsormidium strain transcriptomes and annotated transcripts for GMIs.

The reviewed MS is important for better understanding the ecology of microorganisms. I recommend it for publication after some corrections.

Suggestions for authors.

Major suggestions:

  1. Add information about the ecology of the habitat (section 2.1).
  2. Add a description of light microscopy analysis to the “Materials and Methods” section.
  3. Improve the resolution of the figures 3, 4-6, or increase them.
  4. Please add at least 20 recent (published in the years 2020-2025) references in the reference list. According to MDPI journal requirements, most newly published papers should be mentioned in the papers. I was able to find only 6 recent publications in your list.

Minor suggestions:

  1. Keywords: Replace “Antarctica” and “Klebsormidium,” because you mentioned them in the title. Include “gene marker indices (GMIs)” in the keywords.
  2. Figure 1: Try to choose another color or style for labeling sites on pictures, because the yellow color is very poorly visible.
  3. Lines 58, 139, and 141: Is it possible to delete parentheses?
  4. Line 189: Delete the dot in the title of section 3.1.

Author Response

I enjoy reviewing the MS about the detection of environmental stress in Klebsormidium using a metatranscriptomic approach. I agree with the authors that Klebsormidium is one of the most widely distributed genera with high tolerance to extreme ecological factors. I am sure that the reviewed article will be of interest to a wide range of scientists. In the study, the authors used proper molecular-genetic and statistical methods. All parts of the MS are clear, and the data are presented logically. It is necessary to note that the paper is well edited. The results are illustrated by clear figures. The conclusions of the paper are based on the results. The supplementary file contains useful information about Klebsormidium strain transcriptomes and annotated transcripts for GMIs.

The reviewed MS is important for better understanding the ecology of microorganisms. I recommend it for publication after some corrections.

            We thank the reviewer for his kind words. We really appreciate it.

Suggestions for authors.

Major suggestions:

  1. Add information about the ecology of the habitat (section 2.1).

We added a sentence on the habitats at Byers Peninsula.

  1. Add a description of light microscopy analysis to the “Materials and Methods” section.

Done.

  1. Improve the resolution of the figures 3, 4-6, or increase them.

New versions for the mentioned Figures have been added to the manuscript.

  1. Please add at least 20 recent (published in the years 2020-2025) references in the reference list. According to MDPI journal requirements, most newly published papers should be mentioned in the papers. I was able to find only 6 recent publications in your list.

The paper cites work from 2000 onward, and much of the cited work is from 2010 onward. We believe that we have adequately reflected the current knowledge on this topic. For this reason, we did not add references just because they are from 2020 to 2025.

Minor suggestions:

  1. Keywords: Replace “Antarctica” and “Klebsormidium,” because you mentioned them in the title. Include “gene marker indices (GMIs)” in the keywords.

Done

  1. Figure 1: Try to choose another color or style for labeling sites on pictures, because the yellow color is very poorly visible.

Done

  1. Lines 58, 139, and 141: Is it possible to delete parentheses?

Done

  1. Line 189: Delete the dot in the title of section 3.1.

Done

Reviewer 2 Report

Comments and Suggestions for Authors

Well documented and tested.  Just a thought: the organisms that live in these environments, such as the Antarctic, have the genetic bandwidth to adapt and survive in these locations.  This is manifest by the transcriptomes/proteins they produce for tolerating these conditions, which we as humans consider stress.

Author Response

Well documented and tested.  Just a thought: the organisms that live in these environments, such as the Antarctic, have the genetic bandwidth to adapt and survive in these locations.  This is manifest by the transcriptomes/proteins they produce for tolerating these conditions, which we as humans consider stress.

We thank the reviewer for the kind words.

Reviewer 3 Report

Comments and Suggestions for Authors

The manuscript by Palaniappan, Pushkareva and Becker have undertake a study of the metatranscriptome of a species of soil crust algae from the Antarctic. The rationale for the study was to determine whether gene markers for cold and dessication stress measured in situ differed from laboratory studies. The authors highlight the novelty of this work.

I am no expert of metatranscriptomics so I trust the other reviewer(s) have that appropriate expertise.

The aims of the study are admirable in that they draw attention to the potentially discrepancies between lab studies and data collected in the field - this is worthwhile and I hope that other researchers will pick up on this and further test this observation.

I only have a few observations that I would like to highlight - these are more of presentational in manner.

Firstly, in Fig 1 I found it difficult to read the yellow text indicating sites 1-3. I would ask that these labels be mare clearer.

On line 114, the authors write: "Due to technical constraints, only three and four replicates from sites 1 and 3, respectively,  were submitted for sequencing." Firstly, do you mean three and four replicates in total or do you mean replicates three and four? Please clarify. Also, please clarify what the "...technical constraints..." were, and why they arose. 

The sentence that starts on line 123 and ends on line 128, you are missing a bracket within this.

I found Figs 5 and 6 rather hard to read - too faint. The presentational quality needs to be improved.

Author Response

The manuscript by Palaniappan, Pushkareva and Becker have undertaken a study of the metatranscriptome of a species of soil crust algae from the Antarctic. The rationale for the study was to determine whether gene markers for cold and dessication stress measured in situ differed from laboratory studies. The authors highlight the novelty of this work.

I am no expert of metatranscriptomics so I trust the other reviewer(s) have that appropriate expertise.

The aims of the study are admirable in that they draw attention to the potentially discrepancies between lab studies and data collected in the field - this is worthwhile and I hope that other researchers will pick up on this and further test this observation.

            Thank you for the kind words.

I only have a few observations that I would like to highlight - these are more of presentational in manner.

Firstly, in Fig 1 I found it difficult to read the yellow text indicating sites 1-3. I would ask that these labels be mare clearer.

            We changed the color to yellow to make it easier to read.

On line 114, the authors write: "Due to technical constraints, only three and four replicates from sites 1 and 3, respectively,  were submitted for sequencing." Firstly, do you mean three and four replicates in total or do you mean replicates three and four? Please clarify. Also, please clarify what the "...technical constraints..." were, and why they arose.

5 replicates were collected but for site 1 only 3 replicates and for site 3 4 replicates gave total RNA of suitable quality. For this reason 3 replicates were sequened for site 1 and 4 replicates for site 3. We modified the text accordingly

The sentence that starts on line 123 and ends on line 128, you are missing a bracket within this.

            We added brackets to make this sentence better understandable.

I found Figs 5 and 6 rather hard to read - too faint. The presentational quality needs to be improved.

New, easier-to-read versions of Figures 5 and 6 are now included.
